# Near-Complete Genome of SARS-CoV-2 Delta (AY.3) Variant Identified in a Dog in Kansas, USA

**DOI:** 10.3390/v13102104

**Published:** 2021-10-19

**Authors:** Tyler Doerksen, Andrea Lu, Lance Noll, Kelli Almes, Jianfa Bai, David Upchurch, Rachel Palinski

**Affiliations:** 1Veterinary Diagnostic Laboratory, College of Veterinary Medicine, Kansas State University, 2005 Research Park, Manhattan, KS 66502, USA; tdoerks@vet.k-state.edu (T.D.); andrealu@vet.k-state.edu (A.L.); lwnoll@vet.k-state.edu (L.N.); kalmes@vet.k-state.edu (K.A.); jbai@vet.k-state.edu (J.B.); 2Veterinary Health Center, College of Veterinary Medicine, Kansas State University, 1800 Denison Ave, Manhattan, KS 66502, USA; upchuda@vet.k-state.edu

**Keywords:** SARS-CoV-2, COVID-19, canine, dog, delta variant, AY.3

## Abstract

Severe acute respiratory syndrome coronavirus 2 (SARS-CoV-2) descriptions of infection and transmission have been increasing in companion animals in the past year. Although canine susceptibility is generally considered low, their role in the COVID-19 disease cycle remains unknown. In this study, we detected and sequenced a delta variant (AY.3) from a 12-year-old Collie living with owners that previously tested positive for SARS-CoV-2. It is unclear if the dogs’ symptoms were related to SARS-CoV-2 infection or underlying conditions. The whole genome sequence obtained from the dog sample had several unique consensus level changes not previously identified in a SARS-CoV-2 genome that may play a role in the rapid adaptation from humans to dogs. Within the spike coding region, 5/7 of the subconsensus variants identified in the dog sequence were also identified in the closest in-house human reference case. Taken together, the whole genome sequence, and phylogenetic and subconsensus variant analyses indicate the virus infecting the animal originated from a local outbreak cluster. The results of these analyses emphasize the importance of rapid detection and characterization of SARS-CoV-2 variants of concern in companion animals.

## 1. Introduction

In late 2019, a novel severe acute respiratory syndrome coronavirus (SARS-CoV-2) emerged in China and has since become one of the most economically impactful pandemics to date [1]. In the past year, multiple viral variants have been reported, causing various complications including higher rates of transmissibility [2,3,4], immune evasion [5,6,7,8,9], and diagnostic complications [10]. A subset of novel variants has been labeled variants of interest (VOI), on the basis of the preceding factors. VOIs include viruses that are phylogenetically classified into eta (B.1.525), iota (B.1.526), and kappa (B.1.617.1) lineages as well as one unclassified variant (B.1.617.3) [11]. The eta, iota, and kappa lineages were first identified in the United Kingdom, United States, and India, respectively. Following identification, many of these lineages have disseminated worldwide, further perpetuating case rates and clinical symptoms.

The rise of the pandemic and rapid evolution of SARS-CoV-2 started a race to develop a protective and efficacious vaccine, which resulted in development of multiple highly efficacious and safe vaccines [12,13,14]. To date, global vaccination rates are 30.7%, with moderately higher vaccination rates in the United States (54.7%) [15]. Despite the rapid vaccine development and administration efforts, a novel variant (delta lineage; B.1.617.2/AY), capable of evading the vaccinated immune system, originated in India and has since spread globally. Compared to other circulating strains, the delta variants have higher transmissibility and reduced neutralization in post-vaccinated individuals and have consequently been labeled ‘variants of concern’ (VOCs) [4,7]. VOCs are characterized by increased transmissibility, more severe disease outcomes, reduction in neutralization in vaccinated individuals, or failures in diagnostic detection [11]. SARS-CoV-2 variants are classified on the basis of genomic sequencing, specifically related to the Spike (S) protein coding region within the genome. To date, there are four SARS-CoV-2 lineages that are VOCs: alpha (B.1.1.7/Q) [16], beta (B.1.351.1,2,3) [17], delta, and gamma (P.1.) [18]. These lineages were first identified in the United Kingdom, South Africa, India, and Japan/Brazil, respectively.

SARS-CoV-2 is a member of the Betacoronavirus genus within the Coronaviridae family, of which there are five subgenera: Embecovirus, Hibecovirus, Merbecovirus, Nobecovirus, and Sarbecovirus [19]. SARS viruses are the current sole members of the Sarbecovirus subgenus. The SARS-CoV-2 genome is composed of ≈30 kb of positive, single-stranded RNA encoding 12 open reading frames of which the spike (S), membrane (M), nucleocapsid (N), and envelope (E) form the virion [19]. The S protein is the major antigenic protein, binding to the ACE2 host cell receptors for entry and is the focus of V and variant classifications [20,21].

Investigations into potential animal reservoirs of SARS-CoV-2, using the original virus from China, indicated cats and ferrets were permissible to the virus, while dogs, pigs, chickens, and ducks were markedly less susceptible to infection [22,23]. In particular, large cats such as tigers may become infected and demonstrate symptoms of disease [24]. While this is encouraging for pet-owners, studies using the delta variant have shown high levels of RNA and viral shedding from hamsters and Asiatic lions producing mild to moderate clinical signs [25,26]. Markedly, animals infected with the delta variant shed virus for up to 14 days post infection, four days longer than other strains [25]. Serological studies have detected circulating SARS-CoV-2 antibodies in canines, including in 43.9% of dogs in Croatia as well as a single dog in Italy [27,28]. Furthermore, molecular detection and sequencing identified an alpha variant (B.1.1.7) virus from an asymptomatic dog in Spain [29]. Since then, a total of 45 sequences have been obtained from dogs worldwide, 17 of which are in the United States (www.gisaid.org; accessed 25 August 2021). To the authors’ knowledge, no SARS-CoV-2 delta variant sequence from a canine host is currently available.

Here, we document the detection and sequencing of an AY.3 virus from a 12-year-old Collie living with a SARS-CoV-2-infected owner. The animal was admitted to the Kansas State Veterinary Health Center (KSU VHC) for unrelated symptoms and tested positive for SARS-CoV-2 by qRT-PCR two days post-admission. Whole genome sequencing, phylogenetic and subconsenus analyses were performed on the sample confirming transmission from human to animal. These results suggest AY.3 transmission from humans to dogs can occur, potentially affecting the animal at high titers and emphasizes the importance of VOC studies.

## 2. Materials and Methods

### 2.1. Quantitative Reverse Transcription PCR (qRT-PCR) for the Detection of SARS-CoV-2 RNA

A nasal swab was collected from a 12-year-old male Collie and submitted to the Kansas State Veterinary Diagnostic Laboratory (KSVDL) for testing. At KSVDL, RNA was extracted from the sample using the QIAamp Viral RNA Mini kit following manufacturer’s specifications. The Path-ID Multiplex One-Step RT-PCR Kit was used in combination with the CDC N1 target primers (Table 1) to test for SARS-CoV-2 RNA. Prepared master mixes were run on an ABI-7500 machine. Results were reported as Ct values.

### 2.2. SARS-CoV-2 Whole Genome Sequencing

SARS-CoV-2 RNA was amplified from total nucleic acid (see above) using the QIAseq SARS-CoV-2 Primer panel (Qiagen). Primers were removed from the amplicon pools with a 0.9x AMPure XP bead cleanup (Beckman Coulter), diluted, and library prepped using the Nextera XT dual-indexing library prep kit (Illumina). All steps were performed following the manufacturers’ specifications. Libraries were run on an Illumina iSeq 100 using a 300-cycle v2 cartridge.

Raw reads were filtered (at or above Q30) and trimmed for quality and mapped to the Wuhan reference genome (Genbank# NC_045512), and then a consensus sequence was extracted for further analysis. All sequence manipulations were performed in CLC Genomics Workbench v21.0.4 using default parameters (Qiagen).

### 2.3. Subconsensus Variant Analysis

Subconsensus variants were extracted from the read mapping using CLC Genomics Workbench v 21.0.4 and the following parameters: quality score of 30, 20x coverage, forward/reverse ratio of >0.1, a frequency of 5%, and a significance of 5%. A *t*-test was used to compare subconsensus variants in Graphpad Prism v9.

### 2.4. Phylogenetic Analysis

SARS-CoV-2 sequence lineages were determined in Pangolin (https://pangolin.cog-uk.io/) and the Nextclade tool (https://clades.nextstrain.org/; accessed 25 August 2021) assigned clades and were used to perform the phylogenetic analysis. Phylogenetic analysis was performed on delta variant in-house sequences (Appendix A) obtained from Manhattan, KS. Sequences with >1% Ns in the S gene were excluded from the analysis, resulting in 30 full genomes in the dataset. The sequence alignment was created in mafft v 7.475 [30] using default parameters and was used to build a maximum likelihood tree in FastTree V2.1 [31] with a general time reversible (GTR) model. The best-fit model was determined to be GTR in Mega-X [32]. Trees were formatted in iTOL (https://itol.embl.de/; accessed 25 August 2021).

## 3. Results

### 3.1. Case Description

A 12-year-old male Collie was admitted to KSU VHC for collapsing following travel. The dog was diagnosed with a hemoabdomen secondary to a bleeding splenic mass after an abdominal FAST scan and abdomincentesis. Thoracic radiographs were also performed and showed a multilobar alveolar pulmonary pattern with mediastinal shift. Differentials included multilobar pneumonia, atelectasis, or multifocal pulmonary hemorrhage. A splenectomy was performed the day following admission, while in surgery, multifocal hepatic nodules were noted as well as a 5 × 4 cm mass on the left medial liver lobe. A complete liver lobectomy was performed to remove the mass. Pulmonary oximetry performed after surgery was between 88 and 90%, indicating poor perfusion, and therefore the dog was housed in an oxygen cage overnight. Thoracic radiographs were performed the next day, which showed pulmonary changes consistent with progressive aspiration pneumonia and mild cylindrical bronchiectasis. The animal was released 5 days after admission without the requirement for supplemental oxygen. The dog was found dead two days after discharge. No postmortem examination or ancillary testing was performed. The owner had tested positive for SARS-CoV-2 prior to the dog’s admission.

### 3.2. Molecular Detection and Sequencing

The nasal swab sample collected from the dog following admission to the KSU VHC was tested and confirmed SARS-CoV-2-positive with a Ct of 12.17 two days after admission. The positive nucleic acid was prepped for whole genome sequencing immediately following qRT-PCR confirmation. A total of 1,458,751 reads mapped to the reference genome (Genbank# NC_045512) resulting in an average coverage of 6896x and was designated hCoV-19/dog/USA/KS-8074/2021 (GISAID # EPI_ISL_4253995). A complete SARS-CoV-2 coding region and partial 5′ and 3′UTRs was extracted from the deep sequencing data. The genome is 29,884 nucleotides in length with a GC content of 38.0% and encodes 12 open reading frames of the expected sizes. The genome is 99.96% identical to the next closest genome (an in-house sample, hCoV-19/USA/KS-KSU-2046/2021, GISAID# EPI_ISL_3693315) equating to 8 nucleotide differences. Of these nucleotide differences (4 in ORF1; 2 in S; 1 in M; 1 in N), three nonsynonymous (NS) changes occurred (1 in ORF1; 1 in S; 1 in N), none of which have been seen in delta viruses previously (Table 2).

### 3.3. Subconsensus Analysis

Subconsensus variants occurred in the ORF1ab and S coding regions only (Figure 1). In total, 71 subconsensus variants were identified, 8 were found in the S coding region (7 NS, and 1 Synonymous (S)) and 63 in the ORF1ab coding region (47NS and 16S). Of the spike nonsynonymous variants, 6/7 have never been identified at the consensus level using CoV-GLUE. The single spike variant identified in previous sequences was present at consensus level in the Wuhan reference genome. Interestingly, 5/7 of the NS variants identified in the S gene were also identified above 10% frequency in hCoV-19/USA/KS-KSU-2046/2021 (Figure 1). This trend was not reflected in the ORF 1ab data as only 1/47 NS subconsensus variants were also present in hCoV-19/USA/KS-KSU-2046/2021.

Subconsensus variants for the Kansas (human) batch of samples were analyzed for trends differing from those that were found in the hCoV-19/dog/USA/KS-8074/2021 sample. Interestingly, the nsp13 coding region within ORF 1b had a significantly greater (*p* = 0.001) number of subconsensus variants than the Kansas (human) group (37 versus an average of 4.1). No other gene-coding region had a significantly different number of variants than the dog sequence.

### 3.4. Phylogenetic Analysis and Comparison to Known Sequences

A phylogenetic tree was constructed from hCoV-19/dog/USA/KS-8074/2021 and the Kansas (human) delta variant references using the Nextclade bioinformatics pipeline (locally) (Figure 2). The resulting analysis confirmed hCoV-19/dog/USA/KS-8074/2021 is most similar to our in-house sequence, hCoV-19/USA/KS-KSU-2046/2021, previously collected from a 21-year-old woman over a month prior. All Kansas sequences (human and dog) had 31 total consensus mutations in common as compared to the reference genome from Wuhan (5′UTR, 2; ORF1ab, 13; S, 6; 3a, 1; E, 1; 7ab, 2; 8, 1; N, 4; 3′UTR, 1) (Appendix A). Interestingly, the Kansas (human) sequences did not form a monophyletic clade reflecting infection from multiple sources. Kansas (human) sequences are interspersed in clades containing sequences from multiple states, suggesting the Kansas (human) sequences are a result of travel-related transmission events. The cluster of Kansas cases surrounding the dog sequence reflect a potential transmission cluster within the city.

## 4. Discussion

To our knowledge, this is the first report of a dog infected with a SARS-CoV-2 delta variant. SARS-CoV-2 transmission from humans to pets has been documented, although infrequently. In this case, it was unclear as to whether the clinical signs post-operation were, in part, a result of the acute SARS-CoV-2 infection or whether the virus contributed to the animal’s death. Although canine susceptibility to SARS-CoV-2 alpha variants has been determined to be generally low [23,33], infectivity of the delta variant has not been tested in dogs. The low Ct value suggests an acute infection that could be a result of a compromised immune system associated with the hepatic and splenic masses, despite the congruence of some of the clinical signs to COVID-19 infection such as pneumonia, bronchiectasis, and lack of oxygenation. Adequate determination is no longer possible.

The high RNA load in the nasal swab suggests the dog may have been shedding virus, causing a risk of infection to surrounding susceptible individuals. To date, a universal protocol for quarantining SARS-CoV-2-positive animals has not been established. The lack of available guidelines is a risk to not only the health of the admitted animals but to clinic staff. Our results indicate steps should be taken to protect the health and environment of veterinary clinics to limit the potential for SARS-CoV-2 transmission.

The patient history coupled with our results confirm SARS-CoV-2 transmission from owner to pet. The owner tested positive prior to admission of the animal to KSU VHC. Transmission from infected owners to pets has been documented for the B.1.1.7 variant in the United States recently in a similar manner as we describe [33]. Phylogenetic and subconsensus analyses also indicate transmission occurred from a cluster of human cases in Manhattan starting more than a month prior to the dog’s positive SARS-CoV-2 test. This is particularly apparent in the S coding region as 5/7 of the NS variants identified in the gene-coding region were also found in the most similar in-house sequence collected over a month prior. Coupled with the high RNA load and potential to shed infectious SARS-CoV-2 virus, this confirms the high transmissibility of this viral lineage.

Coronavirus are known to rapidly evolve following or prior to jumps between species [34,35,36,37]. Our analyses indicate nonsynonymous changes are present in the ORF 1ab, S, and N genes. While these data may not conclude variant involvement in the jump between humans and dogs, the S gene is the target of the host immune system, suggesting the nonsynonymous mutation M1227L may be of particular importance for the delta variant’s rapid adaptation to dogs. The M1227L mutation is located in the S gene on the central hydrophobic portion in the transmembrane domain that functions to stabilize the trimeric structure used for membrane fusion [38,39]. Destabilization of the transmembrane domain within the S gene is associated with reduced viral infectivity and therefore efficiency of infection [40]. This study suggests the M1227L mutation may cause S protein destabilization that could contribute to the transmission of SARS-CoV-2 from humans to dogs.

We speculate that subconsensus variants may also contribute to the transmission between dogs and humans. The non-structural protein 13 in particular had a significantly greater number of subconsensus variants, yet no consensus level variants, in the dog sample as the reference database. This finding is curious considering the helicase is highly conserved across coronavirus family. The significance of this finding is not known.

Monitoring for the emergence of novel variants of SARS-CoV-2 is of critical public health importance. Our findings suggest monitoring may be appropriate and necessary in companion animals as well, especially those that reside in households that have a known SARS-CoV-2 infection. These measures will help to ensure the health of humans and animals in the continued effort to mitigate the risks of SARS-CoV-2.

## Figures and Tables

**Figure 1 viruses-13-02104-f001:**
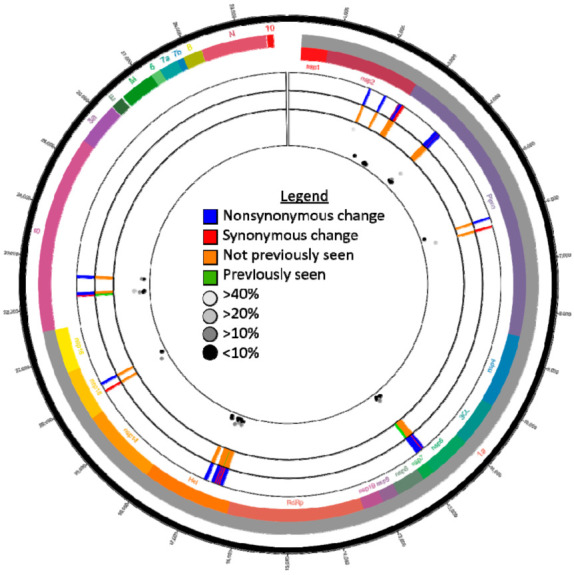
Subconsensus variants identified in hCoV-19/dog/USA/KS-8074/2021 (GISAID # EPI_ISL_4253995) reads using the following conditions: quality score of 30, 20x coverage, forward/reverse ratio of >0.1, a frequency of 5%, and a significance of 5%. The inner scatter plot displays the frequencies of each variant while the two middle the two concentric circles directly surrounding the scatter plot indicate the subconsensus variant sites and metadata taken from CoV-GLUE (http://cov-glue.cvr.gla.ac.uk/; accessed 28 August 2021).

**Figure 2 viruses-13-02104-f002:**
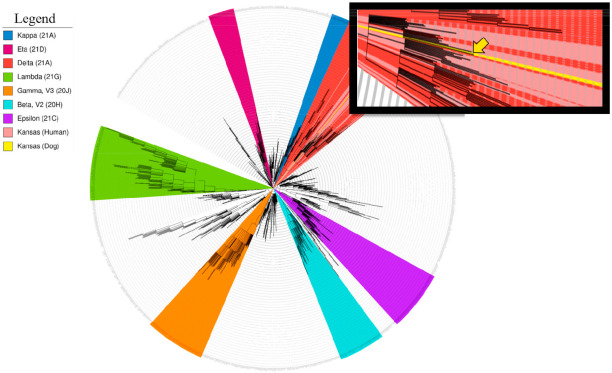
Phylogeny of hCoV-19/dog/USA/KS-8074/2021. The phylogenetic tree was constructed using the Nextclade pipeline, run on a local server using a GTR model. The tree was modified in iTOL. SARS-CoV-2 lineages are illustrated using the following color scheme: kappa, blue; eta, maroon; delta, red; lambda, green; gamma, orange; beta, turquoise; epsilon, purple. The SARS-CoV-2 genomes obtained from people in Manhattan, KS, are indicated in light red. The SARS-CoV-2 dog genome is indicated in yellow in the main figure and with a yellow arrow in the inset. Variants of concern (VOCs) alpha, beta, gamma, epsilon, and delta lineages.

**Table 1 viruses-13-02104-t001:** SARS-CoV-2 qRT-PCR primer sequences.

Primer Name	Primer Sequence
2019-nCoV_N1-F	GACCCCAAAATCAGCGAAAT
2019-nCoV_N1-R	TCTGGTTACTGCCAGTTGAATCTG
2019-nCoV_N1-P	FAM-ACCCCGCATTACGTTTGGTGGACC-IBFQ

**Table 2 viruses-13-02104-t002:** Consensus changes in hCoV-19/dog/USA/KS-8074/2021 (GISAID # EPI_ISL_4253995) as compared to the closest reference hCoV-19/USA/KS-KSU-2046/2021 (GISAID# EPI_ISL_3693315). Nucleotide (NT) and amino acid (AA) changes were identified in ORF 1ab, the spike (S), the matrix (M), and the nucleocapsid (N) coding region only. AA position denoted only if the AA was changed from the reference.

ORF	NT Position	AA Position	AA Change	Identified In
1ab	6754	2153	T > I	Never
1ab	7145			
1ab	7151			
1ab	10,733			
S	23,309			
S	25,272	1227	M > L	Never
M	26,895			
N	29,127	279	P > Q	Alpha lineage

## Data Availability

The whole genome sequences for this study are available on GISAID. Deep sequence data for hCoV-19/dog/USA/KS-8074/2021 is available in NCBI’s Sequence Reads Archive under accession number: PRJNA764863, and in GISAID under accession: GISAID # EPI_ISL_4253995.

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
