# Peer review of "Near-Complete Genome of SARS-CoV-2 Delta (AY.3) Variant Identified in a Dog in Kansas, USA"

_viruses, 2021, doi:10.3390/v13102104_

Round 1

Reviewer 1 Report

This manuscript from Doerksen and colleagues reports on the first case of SARS-CoV-2 delta variant detected in a dog in Kansas. In this study, the authors determined the full-genome sequence of the virus and then conducted an evolutionary analysis of its origin. They also identified some interesting genetic mutations and discussed their implications in adaptation from humans to dogs.

This work is significant as there are some emerging data about potential spread with more severity of SARS-CoV-2 delta variant in companion animals including dogs. Despite more information needed to determine the underlying cause of death for this 12-year-old Collie, it is likely that delta variant may be involved in the disease progression. Interestingly, owners of the dog tested positively for SARS-CoV-2, indicating the likelihood of an in-house transmission event.

The manuscript is developed in a good form. It is well written and easy to follow and comprehend. The main conclusions are supported by presented data and interpretation. Rapid detection and characterization of SARS-CoV-2 and variants of concern is very important as it will be critical to protect people who either work in animal clinic or provide care for their animals.

Minor points:

Figure 2 legend needs to be expanded for clarity by including more information on virus lineages.

Since M1227L mutation in the spike protein is quietly interesting in terms of potential cross-species transmission, it would benefit to the readership if the authors would discuss its precise location in the spike protein and its role in viral entry.

Author Response

We would like to thank all of the reviewers for taking the time to critically review the manuscript. We appreciate how the comments will improve the manuscript. Our specific responses are below:

Reviewer 1: This manuscript from Doerksen and colleagues reports on the first case of SARS-CoV-2 delta variant detected in a dog in Kansas. In this study, the authors determined the full-genome sequence of the virus and then conducted an evolutionary analysis of its origin. They also identified some interesting genetic mutations and discussed their implications in adaptation from humans to dogs. This work is significant as there are some emerging data about potential spread with more severity of SARS-CoV-2 delta variant in companion animals including dogs. Despite more information needed to determine the underlying cause of death for this 12-year-old Collie, it is likely that delta variant may be involved in the disease progression. Interestingly, owners of the dog tested positively for SARS-CoV-2, indicating the likelihood of an in-house transmission event. The manuscript is developed in a good form. It is well written and easy to follow and comprehend. The main conclusions are supported by presented data and interpretation. Rapid detection and characterization of SARS-CoV-2 and variants of concern is very important as it will be critical to protect people who either work in animal clinic or provide care for their animals.

Minor points:

Figure 2 legend needs to be expanded for clarity by including more information on virus lineages. Thank you for this observation. Additional information was included in the description for this figure: SARS-CoV-2 lineages are illustrated using the following color scheme: Kappa, blue; Eta, maroon; Delta, red; Lambda, green; Gamma, Orange; Beta, turquoise; Epsilon, purple. The SARS-CoV-2 genomes obtained from people in Manhat-tan, KS are indicated in light red. The SARS-CoV-2 dog genome is indicated in yellow in the main figure and with a yellow arrow in the inset. Variants of Concern (VOCs) Alpha, Beta, Gamma, Epsilon and Delta Lineages.

Since M1227L mutation in the spike protein is quietly interesting in terms of potential cross-species transmission, it would benefit to the readership if the authors would discuss its precise location in the spike protein and its role in viral entry. Thank you for this recommendation to help strengthen the manuscript. We have added in the below sentences describing the particular location and function of this mutation in the S gene: “The M1227L mutation is located in the S gene on the central hydrophobic portion in the transmembrane domain that functions to stabilize the trimeric structure used for mem-brane fusion [38,39]. Destabilization of the transmembrane domain within the S gene is associated with reduced viral infectivity and therefore efficiency of infection [40]. This study suggests the M1227L mutation may cause S protein destabilization that could con-tribute to the transmission of SARS-CoV-2 from humans to dogs.”

Reviewer 2 Report

My major question is about the reason for testing this canine sample.  Was this part of a standard practice at this clinic? Was this tested based on clinical signs? Was this for research purposes?  The authors included an IRB statement but there is no IACUC approval or client consent listed.

The key part of the third paragraph is the last sentence, which explains what sequence is used for genotyping and classification.  This information needs to be before the discussion of variants.

VOC is used in the abstract but not defined.

Line 97: were reads filtered on quality or just trimmed?

Line 111: is 'Ns' considered a standard abbreviation?

The 'in house' sequences are referenced and inclusion in GISAD is stated, but the reference numbers are not provided (supplemental data) so the analysis cannot be repeated.

A GISAD# is listed in figure 1 but not provided in the text for that sample.

If I understand correctly, the canine sequence is closely related to a human sequence from one month prior and it is closer to this virus than the sequence from the owner.  Does this suggest that the dog got the virus from another source than the owner or is this sequencing error, or ...? This should be discussed.

Line 226: not sure what is meant by 'purpose'.  Could this be 'significance'?

Author Response

We would like to thank all of the reviewers for taking the time to critically review the manuscript. We appreciate how the comments will improve the manuscript. Our specific responses are below:

Reviewer 2:

My major question is about the reason for testing this canine sample. Was this part of a standard practice at this clinic? Was this tested based on clinical signs? Was this for research purposes? The authors included an IRB statement but there is no IACUC approval or client consent listed. Thank you for your inquiry into the purpose of the sample. It is standard practice at the clinic that if the animal is displaying respiratory symptoms and/or has been housed with infected owners, the animal is to be tested prior to surgery. In this case, it was a combination of the two reasons that the animal was tested. The initial test was a routine diagnostic submission. This project is not subject to IACUC approval because the animal was admitted and treated to the Veterinary Health Clinic at Kansas State University which functions as a normal veterinary practice. This study was carried out using routine diagnostic tests.

The key part of the third paragraph is the last sentence, which explains what sequence is used for genotyping and classification. This information needs to be before the discussion of variants. Thank you for this comment. The information was added prior to discussing the variants: SARS-CoV-2 variants are classified based on genomic sequencing, specifically related to the Spike (S) protein coding region within the genome.

VOC is used in the abstract but not defined. Thank you for your comment. “Variants of concern” was added to line 22.

Line 97: were reads filtered on quality or just trimmed? We thank the reviewer for identifying this inconsistency. The reads were filtered for quality (at or above Q30) a subsequently trimmed for quality. This information was added to line 97: “filtered (at or above Q30).”

Line 111: is 'Ns' considered a standard abbreviation? Thank you for this observation. Ns are considered a standard abbreviation for sequences nucleotides with an unknown base.

The 'in house' sequences are referenced and inclusion in GISAD is stated, but the reference numbers are not provided (supplemental data) so the analysis cannot be repeated. Thank you for this observation. We have included a supplemental figure which includes the GISAID accession numbers for the sequences used in the study. The table was also referenced in line 111 of the manuscript.

A GISAD# is listed in figure 1 but not provided in the text for that sample. Thank you for the comment. The GISAID # is mentioned at the first reference to the sequence, in line 139 of the text.

If I understand correctly, the canine sequence is closely related to a human sequence from one month prior and it is closer to this virus than the sequence from the owner. Does this suggest that the dog got the virus from another source than the owner or is this sequencing error, or ...? This should be discussed. Thank you for the question, the suggestion for this finding is that SARS-CoV-2 was transmitted from the owners (infected with the known circulating virus) to the animal. There is no suggestion that the animal was infected by a different source. The samples from the owners were unable to be recovered and sequenced, although would have shown a much better connection. This point is discussed in lines 214-216.

Line 226: not sure what is meant by 'purpose'. Could this be 'significance'? Thank you for your comment. The word ‘purpose’ has been changed to ‘significance.’

Reviewer 3 Report

1. This is an interesting study but I was wondering if any further investigation was done to see if this infected animal transmitted  Delta variants to other animal or human with such quantum of viral load?

2. Can you please describe the  mutations observed in Delta positive sequences in representative sequences provided in the manuscript in Figure 2.

Author Response

We would like to thank all of the reviewers for taking the time to critically review the manuscript. We appreciate how the comments will improve the manuscript. Our specific responses are below:

Reviewer 3:

  1. This is an interesting study but I was wondering if any further investigation was done to see if this infected animal transmitted Delta variants to other animal or human with such quantum of viral load? Thank you for this question. The animal tested positive prior to surgery and was therefore, isolated in the hospital to eliminate the possibility of transmission. Additionally, the animal died soon after release and was not exposed to another animal in that time, to our knowledge. The animal handlers, knowing that the animal was positive, also took extra precautions to limit transmission to themselves or others in this time.
  2. 2. Can you please describe the mutations observed in Delta positive sequences in representative sequences provided in the manuscript in Figure 2.Thank you for the recommendation. This sentence was added to the manuscript, “All Kansas sequences (human and dog) had 31 total consensus mutations in common as compared to the reference genome from Wuhan (5’UTR, 2; ORF1ab, 13; S, 6; 3a, 1; E, 1; 7ab, 2; 8, 1; N, 4; 3’UTR, 1) (Supplementary Table 2).” Additionally, a supplementary table (2) was added in which shows the specific variants referenced in the manuscript.
